# CAST: CONTRASTIVE ADAPTATION AND DISTILLATION FOR SEMI-SUPERVISED INSTANCE SEGMENTATION

## ABSTRACT

Instance segmentation demands costly per-pixel annotations and computationally expensive models. We introduce CAST, a semi-supervised knowledge distillation (SSKD) framework that compresses pre-trained vision foundation models (VFM) into compact experts using limited labeled and abundant unlabeled data. CAST unfolds in three stages: (1) domain adaptation of the VFM(s) via self-training with contrastive calibration, (2) knowledge transfer through a unified multi-objective loss, and (3) student refinement to mitigate residual pseudo-label bias. Central to CAST is an *instance-aware pixel-wise contrastive loss* that fuses mask and class scores to extract informative negatives and enforce clear inter-instance margins. By maintaining this contrastive signal across both adaptation and distillation, we align teacher and student embeddings and fully leverage unlabeled images. On Cityscapes and ADE20K, our $\approx 11\times$ smaller student improves over its zero-shot VFM teacher(s) by +8.5 and +7.1 AP, surpasses adapted teacher(s) by +3.4 and +1.5 AP, and further outperforms state-of-the-art SSKD methods on both benchmarks.

## 1 INTRODUCTION

Pixel-level instance segmentation is notoriously expensive: annotating detailed masks can take hours per image, and training state-of-the-art detectors often requires hundreds of GPU hours, putting many applications out of reach Cordts et al. (2016); He et al. (2017). Recent advancements in vision foundation models (VFMs) Oquab et al. (2023); Liu et al. (2024); Yuan et al. (2025); Kirillov et al. (2023) have substantially expanded the capabilities of computer vision systems, achieving strong performance across diverse perception benchmarks Awais et al. (2025).

**Motivation.** Despite remarkable achievements, foundation models still cannot serve specific downstream tasks sufficiently well due to two major issues: (1) the heavy computational overhead during deployment making these models impractical for environments with limited resources Xu et al. (2024); and (2) their inherently generic nature, which leads to suboptimal performance on tasks that demand domain specific expertise Sony et al. (2025). The latter stems from foundation models being optimized to perform well across a wide variety of tasks, rather than being finely tuned for the nuanced requirements of specialized applications Bommasani et al. (2021). This challenge is prominent in applications that involve outdoor environments, such as autonomous driving, and indoor settings, such as robotic perception Firoozi et al. (2023). Semi-supervised knowledge distillation (SSKD) for instance segmentation seeks to compress large models into efficient student models by leveraging both limited labeled data and abundant unlabeled images. Current distillation methods either treat VFMs as fixed feature extractors with simple pseudo-labeling or focus on coarse semantic tasks, failing to exploit the rich structure of unlabeled datasets to refine per-pixel predictions. Consequently, adjacent instances remain poorly separated and accuracy degrades sharply under scarce labels. We address these issues by adapting VFMs via self-training to enhance pseudo-label fidelity, and by injecting an instance-aware pixel-wise contrastive loss that leverages unlabeled data to enforce clear inter-instance margins, yielding sharper masks and superior performance in the low-label regime.

**Status quo.** Knowledge distillation has evolved from task-agnostic compression Hinton et al. (2015); Chen et al. (2020b) to adapting VFMs for downstream tasks. For classification and semantic segmentation, Vemulapalli et al. Vemulapalli et al. (2024) distill a VFM matching its output on an

unlabeled transfer set, and SAM-CLIP Wang et al. (2024) fuses CLIP and SAM. However, neither method targets per-pixel instance masks nor exploits dense self-supervision from the unlabeled pool. Pure semi-supervised instance segmentation methods, such as Hu et al. (2023); Berrada et al. (2024) train teachers from scratch, doubling GPU cost, and still produce noisy masks under scarce labels. To our knowledge, no prior work unifies VFM adaptation, unlabeled data-driven pixel-wise refinement, and extreme student compression for instance segmentation.

**Contributions.** We summarize our main contributions as follows:

- We introduce an *instance-aware pixel-wise contrastive loss* that fuses mask and class predictions to drive stronger inter-instance separation, and show how to sample negatives efficiently in an instance centric setting.
- We propose CAST, a SSKD pipeline with three phases: (i) adapting the foundation teacher via self-training with contrastive calibration, (ii) distilling into a compact student using a unified objective that combines supervised, pseudo-label, and pixel-wise contrastive losses, and (iii) supervised fine-tuning to reduce residual bias, unifying supervised, semi-supervised, and self-supervised signals.
- We conduct extensive experiments on Cityscapes and ADE20K, demonstrating that our $\approx 11\times$ smaller student improves over its zero-shot VFM teacher(s) by +8.5 and +7.1 AP, surpasses adapted teacher(s) by +3.4 and +1.5 AP, and further outperforms state-of-the-art semi-supervised instance segmentation methods under the same data splits, with lower training cost.

## 2 RELATED WORK

**Vision Foundation Models.** VFMs Oquab et al. (2023); Liu et al. (2024); Ravi et al. (2024); Yang et al. (2024b); Bochkovskii et al. (2024) have revolutionized computer vision through large scale pre-training. In parallel, recent trends focus on combining VFMs to extend their capabilities Ren et al. (2024); Yuan et al. (2025). While these models excel in open-set recognition and transfer learning, their computational demands yet hinder edge deployment. Recent efforts merge VFMs via distillation: Wang et al. Wang et al. (2024) unify SAM and CLIP via multi-task learning, while Zhang et al. Zhang et al. (2025) distill CLIP and DINOv2 into a compact model with data distillation. We extend these paradigms by leveraging VFMs for instance segmentation, focusing on balancing robustness with computational efficiency.

**Knowledge Distillation in Vision.** Knowledge distillation (KD) has become a ubiquitous technique to transfer knowledge from teachers with high capacity to lightweight students for efficient deployment. Early methods distilled softened logits or intermediate features Hinton et al. (2015) in a task-agnostic way, while later feature-based approaches capture structured spatial cues (e.g., pixel-wise similarity, channel distributions) Rajasegaran et al. (2020); Shu et al. (2021). Modern methods tackle VFMs' scale and complexity: Sun et al. (2023); Yang et al. (2024a) distills VFMs to impart zero-shot and multimodal capabilities, further multi-teacher approaches Jiang et al. (2024); Yang et al. (2025a) combine complementary expertise. Vemulapalli et al. Vemulapalli et al. (2024) adapt a VFM to the target task and then distill on a large unlabeled set for classification and semantic segmentation. Building on these advances in vision knowledge distillation, we posit that a strong teacher (or ensemble of teachers) can effectively guide a lightweight instance segmentation model to high performance. Our approach explicitly integrates semi-supervised learning and pixel-level contrastive signals for instance segmentation, to focus on bridging the gap between rich representation of VFMs and compact, efficient student networks.

A complementary line of work studies contrastive knowledge distillation. CRD Tian et al. (2019) and SEED Fang et al. (2021) reformulate distillation as contrastive alignment of teacher and student representations via memory queues, and CRCD Zhu et al. (2021) enriches these objectives using both feature and gradient relations. Subsequent efforts extend contrastive KD to dense prediction. G-DetKD Yao et al. (2021) contrasts teacher and student ROI features for object detection, CIRKD Yang et al. (2022) introduces pixel-level contrastive distillation via a shared memory bank for semantic segmentation, Af-DCD Fan et al. (2023) reduces memory demand by directly contrasting spatial and channel embeddings, and PCD Huang & Guo (2023) improves correspondence through spatial adaptation. While effective in their respective settings, existing methods do not address the scale of VFMs or the structural heterogeneity between teacher and student models, offer limited support for dense instance-level tasks, and generally assume a shared feature map for a single teacher and student

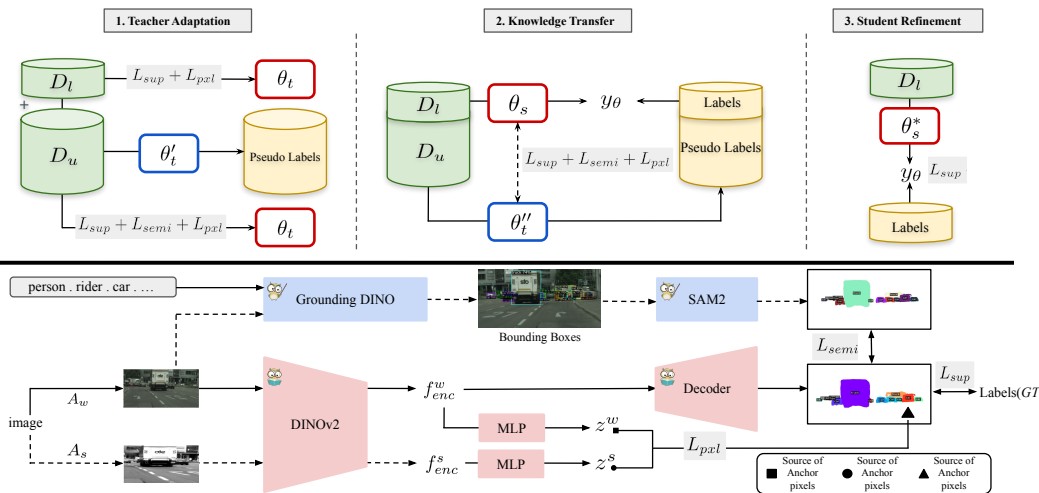

Figure 1: **CAST framework overview. Top:** Three-stage pipeline: (1) adapt a pre-trained VFM teacher to the target domain via self-training with pixel-level contrastive calibration; (2) distill knowledge into a compact student using instance-aware contrastive sampling; (3) fine-tune the student on labeled data to correct residual pseudo-label bias. **Bottom:** Detailed view of stage (2): fused mask and class score maps produce anchor pixels, sampled across weak/strong views to form positive/negative pairs; an MLP projects features for the contrastive loss. Dashed arrows denote no gradient flow; red modules are trainable, blue are frozen.

pair. In contrast, our method does not perform teacher-student contrastive alignment. We employ contrastive learning purely as a self-supervised signal on unlabeled data to enhance the student's structural consistency, enabling effective distillation under our unified KD pipeline.

**Semi-Supervised Learning.** Self-training (or pseudo-labeling) has become a foundational paradigm in semi-supervised learning (SSL), where a model leverages its own predictions with high confidence and iteratively refines itself Xie et al. (2020). This approach has proven effective across vision tasks, improving image classification performance Xie et al. (2020) and boosting object detection accuracy when annotation budgets are tight Liu et al. (2021). To counteract error accumulation from noisy pseudo-labels Tarvainen & Valpola (2017) use exponential moving average of label predictions, or Cascante-Bonilla et al. (2021) employ curriculum labeling schemes that gradually incorporate harder examples. More recent work applies pseudo-labeling for large pre-trained models through targeted finetuning and adaptive pseudo selection strategies Gan & Wei (2024). While many SSL methods focus on classification or detection, several have extended this method to dense prediction tasks Chen et al. (2021a); Yang et al. (2023).

We study self-training with self-supervised contrastive learning and task-specific adaptation. Global contrastive frameworks such as SimCLR Chen et al. (2020a), MoCo Chen et al. (2021b), and their detection extensions Xie et al. (2021a) established the value of large-scale visual discrimination learning. Further per-pixel contrastive approaches Wang et al. (2021); Xie et al. (2021b); Zhong et al. (2021); Wang et al. (2022); Alonso et al. (2021) have shown promise in retaining spatial sensitivity though they yet conflate pixels from different instances of the same class. We extend these advances by synergizing self-training and self-supervised contrastive learning, and introduce a novel instance-aware negative sampling strategy designed specifically for the demands of instance segmentation.

## 3 METHOD

### 3.1 OVERVIEW

In semi-supervised settings, we are given a small labeled set and a substantially larger unlabeled pool:

$$\mathcal{D}^l = \left\{ (x_i^l, y_i^l) \right\}_{i=1}^{N_l} \quad \text{and} \quad \mathcal{D}^u = \left\{ x_i^u \right\}_{i=1}^{N_u}, \quad N_u \gg N_l,$$

where each $y_i^l$ consists of binary masks and class labels for every instance. Our goal is to distill knowledge from a large, pretrained VFM into a compact student $f_{\theta_s}$, matching or surpassing the teacher's accuracy with far fewer labels and compute. We propose **CAST**, a three-stage SSKD pipeline that hinges on two core innovations: ❶ *Contrastive Calibration.* We fine-tune a large VFM teacher via self-training, but rather than simple pseudo-labels we inject a pixel-wise contrastive head to sharpen mask boundaries. ❷ *Debiased, Instance-Aware Sampling.* During both adaptation and distillation, we mine hard negatives via a joint mask-/class-probability embedding, focusing repulsion on informative inter-instance pairs tailored for instance segmentation. These two ideas are then realized in three concise stages (see Fig. 1):

1. **Teacher Adaptation.** Self-train the VFM with pseudo-labels *and* pixel-wise contrastive calibration to produce masks specialized to the target domain.
2. **Knowledge Transfer.** Freeze this calibrated teacher and distill into a lightweight student under a unified loss that harmonizes ground truth, pseudo-label, and contrastive terms, guided by our debiased sampling.
3. **Student Refinement.** Fine-tune the student on labeled data to remove residual pseudo-label bias.

Sec. 3.2 formalizes our instance-aware pixel-wise contrastive loss, which is used in both Teacher Adaptation and Knowledge Transfer to enforce intra-instance cohesion and inter-instance separation; Sec. 3.3 then details the three stages of the CAST pipeline.

## 3.2 Pixel-wise Contrastive Loss

Standard supervised and pseudo-label losses enforce correct mask predictions, ignoring pixel-level feature relationships which underutilize unlabeled data and amplify pseudo label noise. We therefore inject a self-supervised pixel-wise contrastive loss as an additional supervisory signal on both labeled and unlabeled images, sharpening feature discrimination and regularizing against noisy labels.

Let $z^{\text{weak}}, z^{\text{strong}} \in R^{B \times N \times D}$ be $\ell_2$-normalized embeddings from two views of each image, where $B$ is the number of images in one mini batch, $N = h \times w$ the number of pixels, and $D$ the embedding dimension. For each pixel $p \in 1, 2, ..., N$ and image index $b \in 1, ..., B$, the corresponding embedding vector is denoted as $z_{b,p} \in \mathbb{R}^D$. We construct the positive pair by sampling the weak and strong embeddings for each pixel. The positive similarity between the two views is

$$s_{b,p}^+ = \langle z_{b,p}^{\text{weak}}, z_{b,p}^{\text{strong}} \rangle / T.$$

Negatives are sampled by our *instance-aware* sampler (§3.2), producing indices $\{(b', q_r)\}_{r=1}^R$ and corresponding similarities $s_{b,p,r}^-$.

$$s_{b,p,r}^- = \langle z_{b,p}^{\text{weak}}, z_{b',q_r}^{\text{strong}} \rangle / T, \quad r = 1, \ldots, R.$$

The pixel-wise contrastive loss is then the standard NT-Xent over all anchors:

$$\mathcal{L}_{\text{pxl}} = -\frac{1}{B\,N} \sum_{b=1}^B \sum_{p=1}^N \log \frac{\exp(s_{b,p}^+)}{\exp(s_{b,p}^+) + \sum_{r=1}^R \exp(s_{b,p,r}^-)}.$$

**Debiased Pixel-Level Negative Sampling.**

To mine true inter-instance pairs without quadratic cost, we derive a per pixel sampling distribution by fusing mask and class probabilities. Let $M \in \mathbb{R}^{B \times K \times H \times W}$, and $L \in \mathbb{R}^{B \times K \times (C+1)}$, be the model's mask and class logits respectively. We first resize $M$ to the feature resolution ($h \times w$) and then normalize logits to probability distributions $P_m$ and $P_c$ via softmax along instance and class dimensions respectively.

For each pixel index $(b, p)$ to find the aggregated class vote, we compute Expected class distribution $F_c$. Further to avoid losing encoded instance ids over aggregation in expected class distribution we form a joint "pseudo probability" embedding by concatenation the mask distribution and class cues in a single vector which gives a richer embedding letting the contrastive head learn arbitrary interactions between mask and class. leading to pseudo probability map be $y[b, p]$.

$$F_c[b, p, c] = \sum_{k=1}^K P_m[b, k, p]\, P_c[b, k, c], \quad y[b, p] = \begin{bmatrix} P_m[b, 1:K, p] \\ F_c[b, p, 1:C+1] \end{bmatrix} \in R^{K+(C+1)}.$$

We score any two pixels $(b, p) \neq (b', q)$ by $\tilde{y}$ being $\ell_2$-normalized vector of pseudo probability map.

$$s^{\mathrm{deb}}\big((b,p), (b',q)\big) = \max\big(0,\ 1 - \langle \tilde{y}[b,p],\ \tilde{y}[b',q]\rangle\big),$$

We draw $R$ negatives $\{q_r\}$ for each anchor $(b, p)$ by sampling proportional to $s^{\mathrm{deb}}$, and then plug these into the NT-Xent denominator of $\mathcal{L}_{\mathrm{pxl}}$.

**Theoretical Insight.** To give a formal rationale for augmenting our pixel-wise contrastive loss, we show that even under a mild negative sampling guarantee, each gradient step on our contrastive term provably increases the expected inter-instance margin.

**Assumption 3.1** (Negative Sampling Guarantee). *When sampling a negative under our instance aware scheme, the probability it originates from a different instance is at least $p > 0.5$, where $p$ can be estimated empirically (see Sec. 4.3).*

**Proposition 3.1** (Expected Margin Growth). *Under Assumption 3.1, one gradient update on $\mathcal{L}_{\mathrm{pxl}}$ increases the expected inter-instance margin $\Delta_{\mathrm{emp}}$ by*

$$\varepsilon = \Theta(p\,\lambda_{\mathrm{pxl}}) > 0.$$

*This expectation holds even when pseudo-labels are imperfect, provided negatives are sampled using our instance aware strategy.*

In practice, raising $\lambda_{\mathrm{pxl}}$ enhances margin growth but also increases training cost. If $\lambda_{\mathrm{pxl}}$ is too large, it can overemphasize inter-instance separation at the expense of intra-instance cohesion. We validate this effect in Sec. 4.3 and provide a proof sketch in Appendix C.

## 3.3 CAST FRAMEWORK

We cast teacher adaptation, student distillation and student refinement as special cases of the same objective with three terms. Let

$$\mathcal{J}(\theta;\, \mathcal{D}^l, \mathcal{D}^u;\, \lambda_{\mathrm{semi}}, \lambda_{\mathrm{pxl}}) = \underbrace{\frac{1}{N_l}\sum_{i=1}^{N_l} \ell\big(f_\theta(x_i^l), y_i^l\big)}_{\mathcal{L}_{\mathrm{sup}}} + \lambda_{\mathrm{semi}} \underbrace{\frac{1}{N_u}\sum_{j=1}^{N_u} \ell\big(f_\theta(x_j^u), \hat{y}_j^u\big)}_{\mathcal{L}_{\mathrm{semi}}} + \lambda_{\mathrm{pxl}}\, \mathcal{L}_{\mathrm{pxl}}\big(\theta;\, \mathcal{D}^l \cup \mathcal{D}^u\big),$$

where $\mathcal{D}^u = \varnothing$ makes the middle term zero.

**Teacher adaptation.** Starting from pretrained weights $\theta_T^0$, we first fine-tune on the labeled set $\mathcal{D}^l$:

$$\theta_T' = \arg\min_\theta\ \mathcal{J}\big(\theta;\, \mathcal{D}^l, \varnothing;\, 0, \lambda_{\mathrm{pxl}}\big).$$

We then generate pseudo-labels $\hat{y}_j^u = f_{\theta_T'}(x_j^u)$, reset to $\theta_T^0$ and fine-tune on $\mathcal{D}^l \cup \{(x_j^u, \hat{y}_j^u)\}$:

$$\theta_T'' = \arg\min_\theta\ \mathcal{J}\big(\theta;\, \mathcal{D}^l, \mathcal{D}^u;\, 1, \lambda_{\mathrm{pxl}}\big).$$

This two-step contrastive calibration yields a specialized teacher whose pseudo-labels are both accurate and spatially consistent for the target domain.

**Knowledge transfer.** With calibrated teacher $\theta_T''$ frozen, student $\theta_s$ is trained via the unified objective:

$$\theta_s^* = \arg\min_{\theta_s}\ \mathcal{J}\big(\theta_s;\, \mathcal{D}^l, \mathcal{D}^u;\, \lambda_{\mathrm{semi}}, \lambda_{\mathrm{pxl}}\big). \tag{1}$$

Here, $\mathcal{L}_{\mathrm{sup}}$ enforces ground truth supervision on $\mathcal{D}^l$, $\mathcal{L}_{\mathrm{semi}}$ distills pseudo-labels from $\mathcal{D}^u$, and $\mathcal{L}_{\mathrm{pxl}}$ imposes our pixel-wise contrastive regularizer across both sets. The coefficients $\lambda_{\mathrm{semi}}$ and $\lambda_{\mathrm{pxl}}$ balance signals, guiding the student to approach teacher's accuracy with far fewer parameters.

**Student Refinement.** Although joint distillation yields a strong initialization, residual pseudo-label noise and contrastive pretext tasks can introduce bias. As a final step, we fine-tune the student on labeled data alone:

$$\theta_s^\dagger = \arg\min_{\theta_s^*}\ \mathcal{J}\big(\theta_s^*;\, \mathcal{D}^l, \varnothing;\, 0, 0\big),$$

This pass removes pseudo-label drift and sharpens decision boundaries for in-domain data.

## 4 EXPERIMENTS

### 4.1 EXPERIMENTAL PROTOCOL

**Datasets.** We evaluate CAST on two standard instance segmentation benchmarks: **Cityscapes** Cordts et al. (2016) contains 2,975 training, 500 validation images of urban street scenes, annotated with 19 semantic categories (8 "thing" classes and 11 "stuff" classes). **ADE20K** Zhou et al. (2019) comprises 20,210 training and 2,000 validation images spanning diverse indoor and outdoor environments, annotated with 150 semantic categories (100 "thing" and 50 "stuff" classes).

**Implementation Details.** All experiments were conducted on Ubuntu 22.04 with Python 3.10 and PyTorch 2.6.0 (CUDA 12.6). Teacher adaptation runs were executed on 2×NVIDIA A100 GPUs, while student training runs used 2×NVIDIA GeForce RTX 4090 GPUs. As a reference, a single fine-tuning run of the teacher (Grounding-DINO) on the supervised Cityscapes split required $\approx 3.5$ GPU hours; a single student training run for this dataset took $\approx 17$ GPU hours.

**Teacher and Student Architectures.** Our teacher is a fused ensemble of Grounding-DINO-Large Liu et al. (2024) and SAM2-L Ravi et al. (2024). Since the SOTA model of Grounding-DINO is closed-source, we use its open-source counterpart mm-Grounding-DINO Zhao et al. (2024). For the student, we pair a DINOv2-S encoder Oquab et al. (2023) with a DPT-S decoder head Ranftl et al. (2021), followed by a lightweight transformer decoder module in the spirit of Mask2Former Cheng et al. (2022). Our choice of the DINOv2+DPT backbone is motivated by the recent successes of "Depth AnythingV2" in monocular depth estimation Yang et al. (2024b) and UniMatchV2 in semantic segmentation Yang et al. (2025b), and aims to facilitate future multimodal fusion work. We evaluate the impact of different student designs in Sec. 4.4, and defer the complete optimizer, learning rate schedules, and other hyperparameters to Appendix B.

### 4.2 MAIN RESULTS

We evaluate a range of knowledge distillation (KD) strategies, ranging from purely supervised to state-of-the-art semi-supervised baselines, and benchmark them against our CAST pipeline. Table 1 reports maskAP and maskAP$_{50}$ on Cityscapes and ADE20K. In the teacher adaptation stage (568M parameters), adding our pixel-level contrastive loss boosts Cityscapes maskAP from 29.7 to 30.5 (+0.8) and maskAP$_{50}$ from 54.9 to 56.6 (+1.7); on ADE20K, maskAP rises from 14.6 to 15.2 (+0.6) and maskAP$_{50}$ from 23.6 to 24.5 (+0.9). These improvements confirm that pixel-wise supervision sharply improves feature discrimination and reduces pseudo-label noise.

In the student distillation stage, our 52M-parameter student (9% of the composite teacher model) achieves 32.2 maskAP and 56.5 maskAP$_{50}$ on Cityscapes with pixel-level loss, outperforming prior SOTA SSKD models. After fine-tuning, the student reaches 33.9 maskAP (+3.4 over the best teacher) and 58.7 maskAP$_{50}$. On ADE20K, it attains 16.1 maskAP and 27.4 maskAP$_{50}$ in the semi-supervised setting, and improves further to 16.7 maskAP (+1.5) and 28.0 maskAP$_{50}$ after fine-tuning, underscoring CAST's robustness across benchmarks. Additional ablations under varied label splits are presented in Section 4.4. To compare efficiency, Figure 2 plots key pipeline efficiency metric on a logarithmic scale for both teacher and student models.

### 4.3 EMPIRICAL VALIDATION

We validate Proposition 3.1 by monitoring the false negative rate (FNR), the fraction of sampled negatives that actually belong to the same instance, and the empirical margin

$$\Delta_{\mathrm{emp}} = \mathrm{NegMean} - \mathrm{PosMean}.$$

Defining $p = 1 - \mathrm{FNR}$ as the success probability of sampling a true negative, Figure 3 shows: the empirical margin every 10 k iterations for $\lambda_{\mathrm{pxl}} \in \{0.01, 0.05, 0.1, 0.2\}$ (left), the raw contrastive loss for $\lambda_{\mathrm{pxl}} = 0.1$ (center), and the false negative rate for $\lambda_{\mathrm{pxl}} = 0.1$ (right, dashed

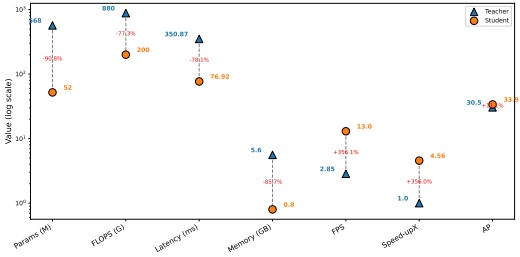

**Figure 2:** Efficiency comparison (log scale).

**Table 1: Main results on Cityscapes and ADE20K** with 10% labeled data. We report teacher adaptation (568M) and student distillation (52M). * denotes adapted methods. Rows in gray are ours.

| Method | Data Regime | Cityscapes | | ADE20K | |
|---|---|---|---|---|---|
| | | maskAP | maskAP$_{50}$ | maskAP | maskAP$_{50}$ |
| *Teacher Adaptation* | | | | | |
| Zero-shot VFM | None (pretrained) | 22.0 | 42.3 | 8.1 | 18.2 |
| Supervised fine-tuning | Labeled only | 28.7 | 53.4 | 14.2 | 23.5 |
| Self-training* Xie et al. (2020) | Labeled+Unlabeled | 29.7 | 54.9 | 14.6 | 23.6 |
| Unbiased Teacher* Liu et al. (2021) | Labeled+Unlabeled | 29.8 | 54.9 | 14.8 | 23.7 |
| CAST (teacher adaptation) | Labeled+Unlabeled | **30.5** | **56.6** | **15.2** | **24.5** |
| *Student Distillation* | | | | | |
| Supervised fine-tuning | Labeled only | 21.1 | 38.7 | 13.9 | 24.2 |
| PAIS Hu et al. (2023) | Labeled+Unlabeled | 22.9 | 44.9 | 10.3 | 18.3 |
| Guided dist. Berrada et al. (2024) | Labeled+Unlabeled | 30.8 | 52.9 | 14.2 | 23.8 |
| Vemulapalli et al.* Vemulapalli et al. (2024) | Unlabeled only | 24.4 | 45.6 | 5.1 | 9.3 |
| CAST (knowledge transfer) | Labeled+Unlabeled | 32.2 | 56.5 | 16.1 | 27.4 |
| CAST (student refinement) | Labeled only | **33.9** | **58.7** | **16.7** | **28.0** |

at $p = 0.5$). Throughout training we observe $p > 0.9$ and a linear increase of $\Delta_{\text{emp}}$ with $\lambda_{\text{pxl}}$, in agreement with Proposition 3.1.

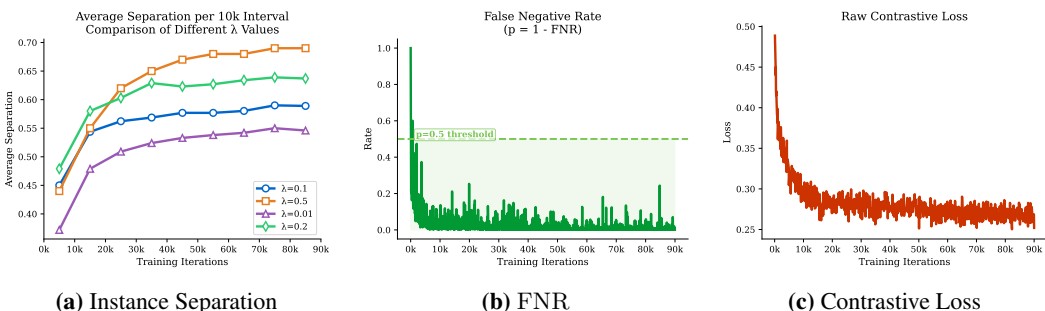

**(a)** Instance Separation       **(b)** FNR       **(c)** Contrastive Loss

Figure 3: **(Left)** Empirical margin (NegMean–PosMean) every 10k iterations for various $\lambda_{\text{pxl}}$. **(Center)** False negative rate (FNR) for $\lambda_{\text{pxl}} = 0.1$, dashed at $p = 0.5$. **(Right)** Contrastive loss for $\lambda_{\text{pxl}} = 0.1$.

### 4.4 ABLATION STUDIES

We perform a series of ablation experiments to isolate the contributions of each component in the CAST pipeline. These include analyses of loss functions, training stages, negative sampling strategies, hyperparameters, and student architecture choices.

**Impact of Loss Components.** During distillation, the objective combines three terms: supervised loss ($\mathcal{L}_{\text{sup}}$), semi-supervised pseudo-label loss ($\mathcal{L}_{\text{semi}}$), and pixel-level self-supervised contrastive loss ($\mathcal{L}_{\text{pxl}}$). Table 2a shows that adding $\mathcal{L}_{\text{semi}}$ improves student performance from 21.1 to 30.7 maskAP, while further including $\mathcal{L}_{\text{pxl}}$ yields the best result of 32.2 maskAP, confirming complementary benefit.

**Table 2:** Ablations on Cityscapes (10% labels). Left: effect of loss terms. Right: effect of CAST stages.

| Method | $\mathcal{L}_{\text{sup}}$ | $\mathcal{L}_{\text{semi}}$ | $\mathcal{L}_{\text{pxl}}$ | Teacher | Student |
|---|---|---|---|---|---|
| (a) Sup. only | ✓ | | | 28.7 | 21.1 |
| (b) + Pseudo | ✓ | ✓ | | 29.7 | 30.7 |
| (c) + Pixel loss | ✓ | | ✓ | 29.6 | 27.5 |
| (d) (b)+(c) | ✓ | ✓ | ✓ | **30.5** | **32.2** |

**(a)** Loss ablation

| Variant | Teacher Adapt. | Distill. | Student FT | maskAP |
|---|---|---|---|---|
| Full CAST | ✓ | ✓ | ✓ | 33.9 |
| No Student FT | ✓ | ✓ | | 32.2 |
| No Teacher Adapt. | | ✓ | ✓ | 25.7 |
| Distillation Only | | ✓ | | 23.8 |
| No Distill. (Sup.) | | | ✓ | 21.1 |

**(b)** Stage ablation

**Impact of Training Stages.** Beyond the contribution of individual loss terms, we further ablate each stage of CAST to justify their necessity. Table 2b shows results on Cityscapes (10% labels), where we drop exactly one stage at a time.

The supervised baseline achieves 21.1 maskAP. Adding distillation alone improves this to 23.8 (+2.7), and further adding student fine-tuning raises it to 32.2 (+8.4). Without teacher adaptation, performance drops to 25.7, underscoring the need to align the teacher with the target domain. The full three-stage CAST pipeline achieves best result of 33.9 maskAP, a +12.8 improvement over baseline.

**Ablation of Negative Sampling via Various Probability Maps.** To validate our negative sampling strategy in the pixel-level contrastive loss, Table 3a compares four sampling methods: **Uniform:** negatives sampled uniformly across the image; **Mask-Only:** The probability map is derived solely from mask predictions, with class probabilities assumed to be uniform. **Class-Only:** The map is generated only from class predictions, assuming a uniform spatial distribution for the mask. **Fusion:** Combining both mask and class predictions. The fusion strategy achieves the best results, with 32.2 maskAP and 56.5 $AP_{50}$.

| **(a)** Negative Sampling Strategies | | |
|---|---|---|
| Method | maskAP (%) | maskAP$_{50}$ (%) |
| Uniform | 29.4 | 50.2 |
| Mask-Only | 30.6 | 55.0 |
| Class-Only | 31.1 | 55.3 |
| Fusion | **32.2** | **56.5** |

**(b)** Schematic of Sampling Distributions

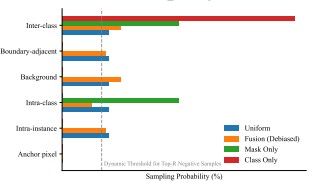

Table 3: **Ablation of Negative Sampling Strategies on Cityscapes.** (a) Quantitative results for uniform, mask-only, class-only, and fusion samplers (maskAP and maskAP$_{50}$). (b) Schematic sketch of the corresponding pixel-level sampling probability distributions.

**Hyperparameter Sensitivity.** We evaluate CAST's sensitivity to three key hyperparameters on Cityscapes: contrastive weight $\lambda_{\mathrm{pxl}}$, negatives per anchor $K$, and temperature $T$, by measuring both teacher and student maskAP (%) and maskAP$_{50}$ (%). Table 4 reports the full sweep. We find that $\lambda_{\mathrm{pxl}} = 0.2$ and $T = 0.2$ consistently maximize performance. For the number of negatives, $K = 256$ offers the best trade-off: although $K = 512$ yields a slight increase in teacher maskAP (30.9 vs. 30.5) and maskAP$_{50}$ (57.1 vs. 56.6), and comparable student metrics, the marginal gains saturate relative to the increased sampling cost. Therefore, we adopt $K = 256$ throughout.

Table 4: **Hyperparameter Ablation on Cityscapes.**

| Model | Metric | Contrastive Loss Weight($\lambda_{\mathrm{pxl}}$) | | | | | Negative Sanples per Anchor($K$) | | | Temperature($T$) | | |
|---|---|---|---|---|---|---|---|---|---|---|---|---|
| | | 0 | 0.01 | 0.1 | **0.2** | 0.5 | 128 | **256** | 512 | 0.1 | **0.2** | 0.4 |
| Teacher | AP | 29.7 | 29.9 | 30.2 | 30.5 | 30.1 | 30.4 | 30.5 | 30.9 | 30.1 | 30.5 | 29.8 |
| | AP$_{50}$ | 55.3 | 55.7 | 56.1 | 56.6 | 56.1 | 56.3 | 56.6 | 57.1 | 55.9 | 56.6 | 55.3 |
| Student | AP | 30.7 | 30.8 | 32.1 | 32.2 | 30.9 | 29.8 | 32.2 | 32.1 | 31.9 | 32.2 | 31.7 |
| | AP$_{50}$ | 54.9 | 55.2 | 56.2 | 56.5 | 55.7 | 55.3 | 56.5 | 56.6 | 56.0 | 56.5 | 55.8 |

**Student Architecture Variants.** We evaluate two design axes for the student model under CAST distillation protocol: (i) the encoder backbone (with a fixed DPT decoder), and (ii) the decoder head (with a fixed DINOv2-S encoder). Table 5 reports accuracy along with parameter counts, on the Cityscapes validation set. The combination of DINOv2-S encoder and DPT head achieves the best accuracy with a compact footprint.

**Scalability with Labeled Fractions.** We evaluate CAST under different fractions of labeled data to assess scalability in semi-supervised settings. Following the protocol in Berrada et al. (2024), we train with 5%, 10%, and 30% labeled splits of Cityscapes. As shown in Table 6, CAST consistently outperforms prior methods across all fractions. At 5% labels, CAST achieves 30.7 AP, far exceeding PAIS (18.0) and Guided Distillation (23.0). At 30% labels, CAST reaches 40.4 AP, surpassing the strongest baseline (37.8 from S[4]M) by +2.6 AP. These results demonstrate that CAST remains effective under scarce supervision while scaling gracefully with additional labeled data. Additional

**Table 5: Architecture Ablations on Cityscapes.** (a) Encoder backbone (fixed DPT decoder). (b) Decoder head (fixed DINOv2-S encoder).

**(a)** Encoder Backbone

| Encoder | maskAP | maskAP$_{50}$ | Params (M) |
|---|---|---|---|
| ResNet50 | 25.5 | 49.3 | 24 |
| SAM2-S | 22.1 | 39.2 | 35 |
| DINOv2-S | **30.7** | **54.9** | 22 |

**(b)** Decoder Head

| Decoder | maskAP | maskAP$_{50}$ | Params (M) |
|---|---|---|---|
| FPN | 28.9 | 52.4 | 18 |
| DPT | **30.7** | **54.9** | 22 |

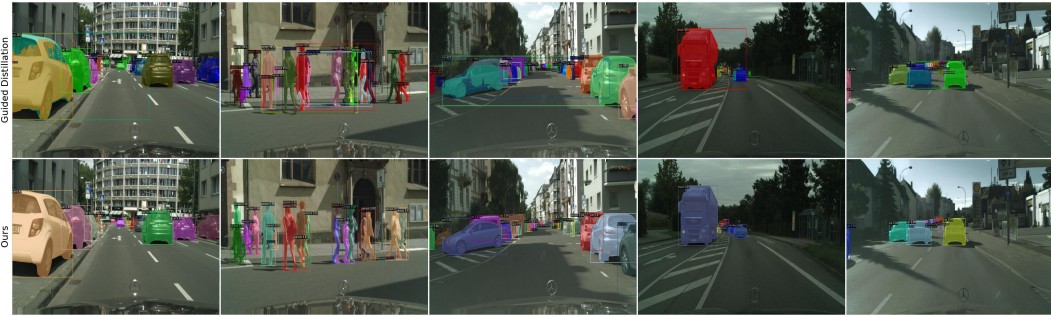

**Figure 4: Qualitative results on Cityscapes.** Guided dist. Berrada et al. (2024) (top) vs. CAST (bottom).

ablations, including teacher adaptation variants, loss formulations, sampling scope, and backbone comparisons, are provided in the supplementary material (Appendix E).

## 5 CONCLUSIONS

We have introduced CAST, a rigorously designed SSKD pipeline that fuses self-training, instance-aware pixel-wise contrastive learning, and final supervised finetuning to compress large VFMs into compact student experts with comparable performance. Empirically, our $\approx 11\times$ smaller student exceeds its adapted teacher by +3.4 maskAP in Cityscapes and +1.5 maskAP in ADE20K, while cutting compute and parameter counts demonstrating that dense contrastive supervision can unlock substantial gains in low-label regimes. Our theoretical analysis further guarantees that our negative sampling scheme provably increases inter-instance margins under mild assumptions. Looking forward, streamlining CAST into a single unified objective, extending its evaluation to diverse domains, and integrating uncertainty quantification will be critical steps toward safe, equitable, and broadly deployable segmentation solutions.

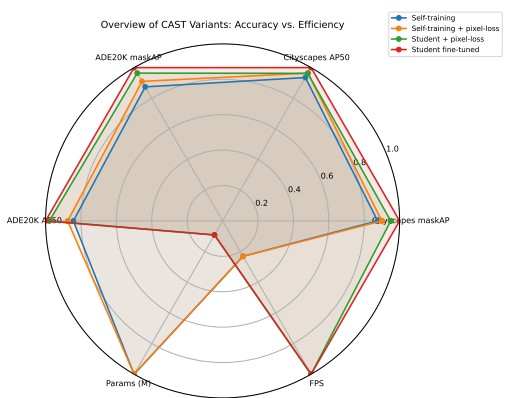

**Figure 5:** Performance–complexity radar chart (normalized).

**Table 6: Scalability across label fractions on Cityscapes.** Results with 5%, 10%, and 30% labeled data.

| Dataset Fraction | Teacher Adapt. | Distillation | CAST (student) | PAIS Hu et al. (2023) | Guided dist. Berrada et al. (2024) | S$^4$M Yoon et al. (2025) |
|---|---|---|---|---|---|---|
| 5% | 29.4 | 29.2 | **30.7** | 18.0 | 23.0 | 30.1 |
| 10% | 30.5 | 32.2 | **33.9** | 22.9 | 30.8 | 33.3 |
| 30% | 33.3 | 38.5 | **40.4** | 32.8 | 35.6 | 37.8 |

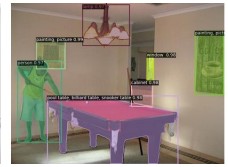 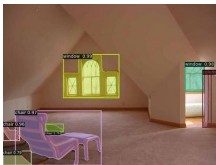 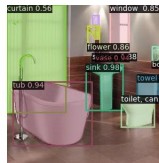 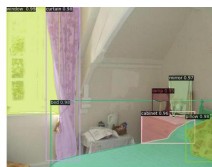

**Figure 6: Qualitative results on ADE20K.**

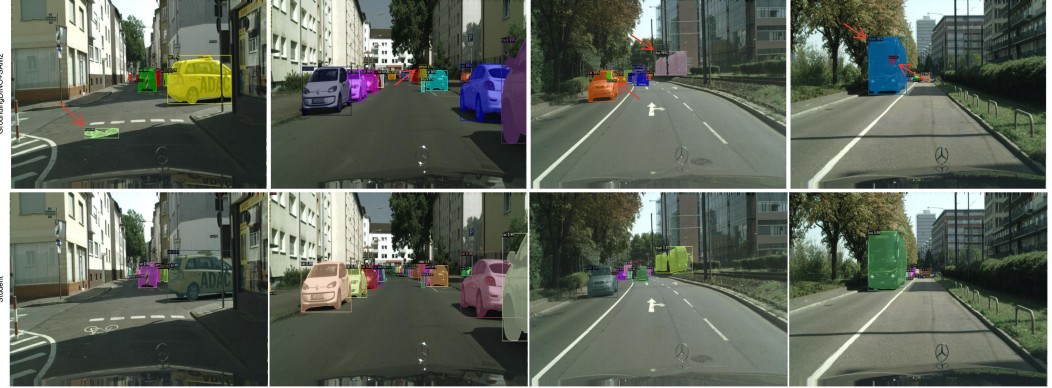

**Figure 7: Qualitative bias reduction in stage-wise distillation.** Top row: pseudo-labels generated by the pretrained teacher. Bottom row: student predictions after distillation and refinement, demonstrating reduced pseudo-label bias and sharper instance boundaries.

## ETHICS STATEMENT

This work does not involve human subjects, private data, or sensitive content. All datasets used are publicly available. Portions of the manuscript were polished using large language model (LLM) for clarity; this use was limited to text editing and did not affect the research process, experiments, or results.

## REPRODUCIBILITY STATEMENT

We provide detailed descriptions of datasets, model architectures, and training procedures to ensure reproducibility. All datasets (Cityscapes, ADE20K) are publicly available, and the 5%, 10%, and 30% labeled splits follow established protocols (Section 4.4). Implementation details, including hyperparameters, software environment, and GPU usage, are reported in Section 4.2, with extensive ablations and sensitivity analyses in Appendix E. To facilitate further research, we will release our code upon the completion of the anonymous review process.

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

## SUPPLEMENTARY MATERIAL

This document provides additional details to support the main paper, including dataset statistics, full hyperparameter settings, formal proof, extended training protocols, and additional ablation studies.

## A    DATASET SPLITS

Table 7 summarizes the datasets used in our experiments. We use a 10% labelled split of Cityscapes' 2 975 training images (298 labeled / 2 677 unlabeled) and a stratified 20% split of ADE20K's 20 210 training images (1 000 labeled / 2 537 unlabeled). Standard validation sets are retained (500 images for Cityscapes, 2 000 for ADE20K). Exact image-ID lists will be released with our code.

**Table 7:** Semi-supervised splits used in our experiments.

| Dataset | # Classes | Labeled / Unlabeled | Validation |
|---------|-----------|---------------------|------------|
| Cityscapes | 8 | 298 / 2 677 | 500 |
| ADE20K | 100 | 1 000 / 2 537 | 2 000 |

## B    HYPERPARAMETERS

Key teacher and student hyperparameters are summarized in Table 8. Results are averages over three independent runs with different random seeds.

**Table 8:** Hyperparameter Settings

| Parameter | Teacher | Student |
|-----------|---------|---------|
| Learning rate | $5.0 \times 10^{-5}$ | Encoder: $5.0 \times 10^{-6}$; Decoder: $5.0 \times 10^{-5}$ |
| Scheduler | Multi-step (milestones at 0.9, 0.95) | PolyLR (power 0.9) |
| Batch size | 4 | 8 |
| Weight decay | 0.01 | 0.05 |
| Contrastive loss weight | 0.2 | 0.2 |
| Pseudo-label threshold | 0.3 | 0.3 |
| Dropout rate | — | 0.1 |
| Gradient clipping | — | $\ell_2$ norm 0.1 |
| Optimizer | AdamW ($\beta1$=0.9, $\beta2$=0.999) | |
| Augmentations | Weak: flip, resize; Strong: random resized crop, jitter, grayscale, blur, | |
| Loss weights (mask / class) | 5 / 2 | |

## C    PROOF SKETCH OF PROPOSITION 3.1

*Proof Sketch.* Let $z_a$, $z^+$ and $\{z_r^-\}_{r=1}^R$ be the unit norm embeddings of an anchor pixel, its positive, and $R$ negatives. Define

$$s^+ = \langle z_a, z^+ \rangle, \qquad s_r^- = \langle z_a, z_r^- \rangle,$$

and the pixel-wise contrastive loss

$$\ell(z_a) = -\log \frac{\exp(s^+)}{\exp(s^+) + \sum_{r=1}^R \exp(s_r^-)}.$$

Let

$$Z = \exp(s^+) + \sum_{r=1}^R \exp(s_r^-), \qquad \alpha_r = \frac{\exp(s_r^-)}{Z}.$$

A straightforward gradient computation gives

$$\nabla_{z_a}\ell = \sum_{r=1}^R \alpha_r \left(z_r^- - z^+\right).$$

Applying one gradient descent step with step size $\lambda_{\text{pxl}}$:

$$z_a' = z_a - \lambda_{\text{pxl}} \nabla_{z_a}\ell = z_a + \lambda_{\text{pxl}} \sum_{r=1}^R \alpha_r \left(z^+ - z_r^-\right).$$

For a randomly chosen negative $z^-$,

$$\Delta s^+ = \langle z_a' - z_a, z^+ \rangle = \lambda_{\mathrm{pxl}} \sum_{r=1}^{R} \alpha_r \big(1 - \langle z_r^-, z^+ \rangle\big),$$

$$\Delta s^- = \langle z_a' - z_a, z^- \rangle = \lambda_{\mathrm{pxl}} \sum_{r=1}^{R} \alpha_r \big(\langle z^+, z^- \rangle - \langle z_r^-, z^- \rangle\big).$$

By Assumption 3.1, each negative embedding $z_r^-$ is inter-instance with probability $p$, in which case $\langle z_r^-, z^+ \rangle \approx 0$, and intra-instance with probability $1 - p$, in which case $\langle z_r^-, z^+ \rangle \approx 1$. Hence

$$\mathbb{E}\big[1 - \langle z_r^-, z^+ \rangle\big] = p \cdot 1 + (1 - p) \cdot 0 = p,$$

and since $\sum_{r=1}^{R} \alpha_r = 1$, it follows that

$$\mathbb{E}[\Delta s^+] = \lambda_{\mathrm{pxl}} \sum_{r=1}^{R} \alpha_r \, \mathbb{E}\big[1 - \langle z_r^-, z^+ \rangle\big] = p \, \lambda_{\mathrm{pxl}}.$$

Meanwhile, every term in $\Delta s^-$ involves an inter-instance inner product, either $\langle z^+, z^- \rangle$ or $\langle z_r^-, z^- \rangle$ each of which vanishes in expectation, so $\mathbb{E}[\Delta s^-] \approx 0$. Therefore

$$\mathbb{E}[\Delta s^+ - \Delta s^-] = p \, \lambda_{\mathrm{pxl}} - 0 = \Theta(p \, \lambda_{\mathrm{pxl}}) = \varepsilon > 0,$$

i.e. one update on $\mathcal{L}_{\mathrm{pxl}}$ increases the expected inter-instance margin by $\varepsilon$. $\qquad\square$

**Remark C.1** (Why $\langle z^+, z^- \rangle \approx 0$ holds). *Under the InfoNCE objective (§3.2), the normalized weights for negative pairs, $\alpha_r = \frac{e^{s_r^-}}{e^{s^+} + \sum_r e^{s_r^-}}$, vanish at convergence, i.e. $\alpha_r \approx 0$. Moreover, in high dimensional embeddings, random unit vectors have inner products concentrating near zero, and contrastive training further pushes these negative similarities into a tight, small magnitude distribution Chen et al. (2020a). Thus it is reasonable to approximate $\langle z^+, z^- \rangle \approx 0$ up to $O(1/\sqrt{D})$ fluctuations.*

## D    MORE TRAINING DETAILS

All teacher models are fine-tuned using 1k iterations on labeled set, followed by 5k iterations in a self-training stage with pseudo-labels. For student models, training on the Cityscapes dataset spans 90k iterations, consistent with prior works, while the mini-ADE20k dataset is trained for 80k iterations. Finally, both datasets undergo an additional supervised fine-tuning phase for 2k iterations.

## E    ADDITIONAL ABLATION STUDIES

### E.1    ABLATION: TEACHER ADAPTATION VARIANTS

Different teacher adaptation strategies impact both teacher and student performance. Specifically, we compare fine-tuning only, self-training, and self-training combined with our proposed contrastive loss.

Table 9: **Teacher Adaptation Ablation.** Teacher/student AP for different adaptation strategies.

| Adaptation Variant | Teacher AP | Student AP | $\Delta$ vs. SOTA |
|---|---|---|---|
| Fine-tuning only | 28.7 | 32.0 | +1.2 |
| Self-training | 29.7 | 32.2 | +1.5 |
| Self-training + Contrastive | **30.5** | **33.9** | **+3.1** |

### E.2    LOSS VARIANT: INFONCE VS. MARGIN HINGE

Replacing our asymmetric InfoNCE (§3.2) with an margin-based hinge loss yields identical maskAP (32.2%) and +0.6 maskAP$_{50}$, at the cost of $1.6\times$ longer training. This evaluates whether enforcing a fixed positive–negative margin can match or improve upon the performance of InfoNCE.

**Table 10: Loss Variant Ablation.** Default InfoNCE vs. margin-based hinge (margin = 0.2).

| Loss Variant | maskAP (%) | maskAP$_{50}$ (%) |
|---|---|---|
| Asymmetric InfoNCE (§3.2) | 32.2 | 56.5 |
| Margin hinge (m = 0.1) | 32.2 | 57.1 |

### E.3 ABLATION: DEBIAS SCORE FORMULATION

We evaluate three instantiations of the debias score function $s^{deb}$ (§3.2):

- **Original** $s^{deb}$: fusion of mask and class confidences (ours).
- $\left(s^{deb}\right)^2$: square each score to amplify the negatives with high confidence.
- $\sqrt{s^{deb}}$: take the square root of each score to temper the bias.

**Table 11: Debias Score Formulation Ablation.**

| Score Variant | maskAP | maskAP$_{50}$ |
|---|---|---|
| Original | 32.2 | 56.5 |
| Squared | 32.0 | 56.3 |
| Square-root | 31.9 | 56.2 |

**Table 12: Teacher Choice Ablation.**

| Model | AP | maskAP$_{50}$ |
|---|---|---|
| Teacher T1 (0-shot) | 22.0 | 42.3 |
| Teacher T2 (adapted) | 30.5 | 56.6 |
| Student under T1 | 23.8 | 42.9 |
| Student under T2 | **32.2** | **56.5** |

### E.4 ABLATION: NEGATIVE SAMPLING SCOPE

We evaluate two negative sampling scopes: (i) sampling only within the current mini batch vs. (ii) sampling from a small memory bank of past pixel embeddings (size 10k). Sampling from a

**Table 13: Sampling Scope Ablation.** Mini batch only vs. memory bank negatives.

| Scope | maskAP (%) | maskAP$_{50}$ (%) |
|---|---|---|
| Mini-batch only | 32.2 | 56.5 |
| Memory bank (10k embeddings) | 32.7 | 57.3 |

memory bank of 10 k embeddings yields a modest performance gain (+0.5 maskAP, +0.8 maskAP$_{50}$) compared to in-batch sampling. However, incurs approximately 2.2× longer training time due to the overhead of maintaining and querying the memory bank.

### E.5 TEACHER CHOICE: ORIGINAL VS. ADAPTED

We compare distilling the student from the original VFM teacher (T1, zero-shot) versus our adapted teacher (T2). As shown in Table 12, using the adapted teacher provides a much stronger signal, yielding a +8.4 AP improvement over the student distilled under T1.

### E.6 EXTENDED BACKBONE COMPARISON

We compare CAST distilled with a DINOv2-S student against Guided Distillation baselines trained with different teacher backbones, including ResNet-50, DINOv2-B, and DINOv2-L.

**Table 14: Extended Backbone Comparison.** CAST vs. Guided Distillation

| Label Fraction | CAST (DINOv2-S) | Guided Dist. (ResNet-50) | Guided Dist. (DINOv2-B) | Guided Dist. (DINOv2-L) |
|---|---|---|---|---|
| 5% | **30.7** | 23.9 | 25.1 | 28.8 |
| 10% | **33.9** | 30.8 | 27.0 | 33.0 |
| 30% | **40.4** | 35.6 | 35.4 | 39.1 |

# F USE OF LLM STATEMENT

We leverage ChatGPT to polish the paper presentation at the sentence level. Specifically, we provided the LLM some of the draft sentences, and asked the LLM if there is a better version of the given sentence

# G ADDITIONAL QUALITATIVE RESULTS

Figure 8 presents additional qualitative examples. The first and third columns show teacher predictions, while the second and fourth columns show the corresponding student predictions.

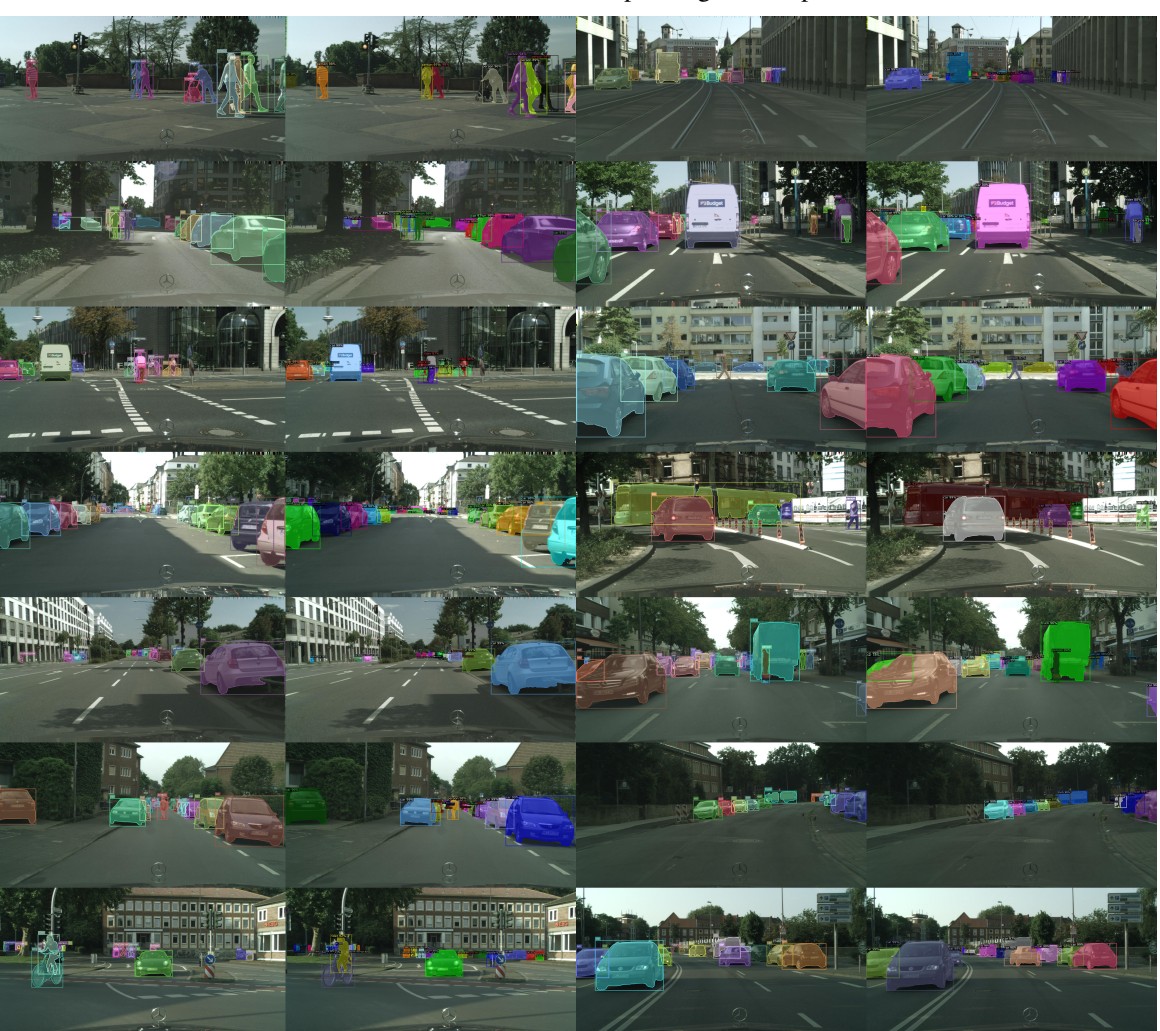

**Figure 8:** Additional qualitative results on the Cityscapes dataset.

