# OpenReview forum: "CAST: Contrastive Adaptation and Distillation for Semi-Supervised Instance Segmentation"
_ICLR.cc/2026/Conference — Submitted to ICLR 2026_

### Official Review · Reviewer_anhQ · 2025-10-28

**Soundness:** 4
**Presentation:** 4
**Contribution:** 4
**Rating:** 6
**Confidence:** 5

**Summary:**

CAST introduces a three-stage Semi-Supervised Knowledge Distillation (SSKD) pipeline that compresses a large Vision Foundation Model (VFM) into a 11× smaller student for instance segmentation. The key novelty is an instance-aware pixel-wise contrastive loss that mines hard negatives by fusing mask + class probabilities; this loss is injected (i) when the teacher is self-trained on unlabeled data and (ii) when the student is distilled.
On Cityscapes and ADE20K with only 10 % labels the student beats its adapted teacher by +3.4 and +1.5 mask AP.

**Strengths:**

1. First work to unify VFM adaptation, dense pixel-level contrastive learning, and extreme compression for instance segmentation.
2. Novel instance-aware negative sampler that fuses mask & class logits to avoid sampling same-instance pixels; backed by a theoretical guarantee that each gradient step increases expected inter-instance margin (Prop. 3.1).
3. Exhaustive ablations (loss terms, stages, sampling, 6 hyper-parameters, 3 label fractions, 2 datasets).
4. Failure modes (pseudo-label bias) are explicitly discussed and mitigated by Stage 3 supervised fine-tuning.
5. Demonstrates that dense contrastive signals can be cheaply harvested from unlabeled images without extra annotations or human-designed rules.

**Weaknesses:**

1. Paper fuses Grounding-DINO + SAM-2 but never distills from single teachers. Readers cannot tell whether gains come from the contrastive loss or from ensembling complementary models.

**Questions:**

I don't have particular questions for this paper.

---

> ### Author Response · Authors · 2025-11-23
>
> We sincerely thank the reviewer for their highly positive evaluation and for recognizing the soundness and contributions of our work. We are particularly grateful that the reviewer highlighted several key strengths, including the novelty of unifying VFM adaptation with dense contrastive learning, the theoretical guarantees provided by our instance-aware negative sampler (Prop. 3.1), and the thoroughness of our ablation studies.
>
>
> Regarding the comment on the fused Grounding-DINO + SAM-2 teacher, we agree this clarification will strengthen the paper. While our ablations already isolate the role of each contribution and training stage, we will further disentangle the individual effects of each teacher to more exclusively attribute their influence separately on downstream performance.

---

> > ### Comment · Reviewer_anhQ · 2025-11-27
> >
> > Thanks for the authors' response. I will maintain my score for this paper.

---

### Official Review · Reviewer_CdV8 · 2025-10-30

**Soundness:** 2
**Presentation:** 2
**Contribution:** 2
**Rating:** 4
**Confidence:** 3

**Summary:**

This paper proposes a semi-supervised knowledge distillation framework for instance segmentation. It proposes an instance-aware pixel-wise contrastive loss to enhance the model's representation capability. Improvements are observed on standard benchmarks like ADE20K and Cityscapes.

**Strengths:**

1. The paper is clearly written and easy to follow.
2. The motivation of using unlabeled data to improve the labor-intensive instance segmentation scenario is good.

**Weaknesses:**

1. The core idea is not new. There are many existing works [1, 2, 3] that explore dense contrastive learning for dense perception tasks, especially three or four years ago when contrastive learning was still very popular.

2. The gain of introducing unlabeled images are marginal. For example, on ADE20K, the supervised-only setting already achieves 23.5 mAP. Using 9x more unlabeled data only improves it by 1.0 mAP.

3. The proposed framework does not exhibit a clear empirical advantage over existing frameworks, such as [4].

[1] Semi-supervised semantic segmentation with pixel-level contrastive learning from a class-wise memory bank, ICCV 2021 \
[2] Sepico: Semantic-guided pixel contrast for domain adaptive semantic segmentation, TPAMI 2022 \
[3] Hunting sparsity: Density-guided contrastive learning for semi-supervised semantic segmentation, CVPR 2023 \
[4] Self-training with noisy student improves imagenet classification, CVPR 2020

**Questions:**

Please see weaknesses.

---

> ### Author Response · Authors · 2025-11-22
>
> We thank the reviewer for recognizing the clarity of our writing and the strength of our motivation. We address all the questions below.
> >**Q1:** The core idea is not new. There are many existing works [1, 2, 3] that explore dense contrastive learning for dense perception tasks, especially three or four years ago when contrastive learning was still very popular.
>
> **A1:**
> Prior works [1-3] primarily employ contrastive learning to enhance knowledge transfer from teacher models to student models in semi-supervised segmentation or domain adaptation settings. [1] uses a memory bank to store class-level features from teacher and student models and applies a positive-only contrastive objective. In contrast, [2] focuses on clustering feature representations rather than leveraging explicit categorical information, extracting supervision from selected anchor and key pairs. Meanwhile, [3] proposes a single-stage domain adaptation framework for semantic segmentation that improves adaptation from a labeled source to an unlabeled target domain by learning class-discriminative pixel embeddings.
>
> In contrast, our CAST framework targets a more realistic and practical scenario: distilling knowledge from Vision Foundation Models (VFMs) into compact student models, under the assumptions that VFMs are readily accessible, unlabeled images are abundant, and labeled images are limited. Our goal is to establish a novel and practical setting that maximizes downstream task performance in the target domain by fully leveraging VFMs.
>
>
> Within this setting, our contributions further investigate: **1.** whether contrastive learning can effectively exploit representations from large amount of unlabeled data in a self-supervised manner to enhance the student’s structural consistency, and **2.** whether this additional self-supervised signal can be combined with supervised and semi-supervised losses [L137].
>
>
> >**Q2:** The gain of introducing unlabeled images are marginal. For example, on ADE20K, the supervised-only setting already achieves 23.5 mAP. Using 9x more unlabeled data only improves it by 1.0 mAP.
>
> **A2:**
> While the raw mAP gain from adding unlabeled images may appear modest in this specific supervised-only baseline (e.g., +1.0 mAP on ADE20K), this interpretation overlooks the nature of the problem setting and the role of unlabeled data in adapting VFMs. There are two key considerations:
>
> 1. VFMs pretrained on billions of images already exhibit strong generalization ability, even before any downstream adaptation. As a result, the gains obtained from additional unlabeled domain data are naturally limited, yet still important for improving recognition of long-tail classes. For example the visual definition of categories such as bus, train, or tram may vary across countries depending on the data that shaped the original pretraining. Figure 7 illustrates this with qualitative examples highlighting confusion between the train and bus classes.
> 2. Unlabeled data becomes substantially more effective when used throughout the CAST pipeline. As clearly shown below:
> | mAP        | cityscapes | ade20k |
> |------------|------------|--------|
> | supervised | 21.1       | 13.9   |
> | **CAST**   | **33.9**   | **16.7** |
>
> > **Q3:** The proposed framework does not exhibit a clear empirical advantage over existing frameworks, such as [4].
>
> **A3:** We respectfully clarify noisy student [4] is a self-training paradigm addressing image-level classification using logit consistency and label-preserving augmentations, while CAST is a knowledge distillation framework for instance segmentation with VFM teachers requiring pixel-wise supervision, object-level separation, and mask boundary reasoning. These two methods operate under fundamentally different task settings, supervision structures, data regimes, and noise formulations, and therefore are not directly comparable.
>
> In the teacher adaptation stage (Table 1, Teacher Adaptation) we compare with strong existing self-training baselines [4,5] not to propose a new self-training paradigm, but to pursue two goals: **1.** We evaluate how different self-training strategies perform when adapting VFMs to the more complex task of instance segmentation. As noted in L295, this setting is not addressed in the original works and therefore required adapting their methods. **2.** We then extend these adapted paradigms with our instance-aware pixel-wise contrastive loss to validate the role of this contrastive signal beyond the student distillation stage. This addition yet consistently improves the adapted teacher across datasets as summarized below.
>
> | Delta maskAP  | cityscapes | ade20k |
> |-----------------------------------|------------|--------|
> | Self-training [4]                 | +1.0 AP    | +0.4 AP |
> | Unbiased Teacher [5]              | +1.1 AP    | +0.6 AP |
> | **CAST (teacher adaptation)**     | **+1.8 AP** | **+1.0 AP** |
>
>
>
> > **Due to space limitations, we provide the reference list in the follow-up comment.**

---

> > ### Author Response · Authors · 2025-11-22
> >
> > ---
> > ### References
> >
> > [1] Semi-supervised semantic segmentation with pixel-level contrastive learning from a class-wise memory bank, ICCV 2021
> >
> > [2] Sepico: Semantic-guided pixel contrast for domain adaptive semantic segmentation, TPAMI 2022
> >
> > [3] Hunting sparsity: Density-guided contrastive learning for semi-supervised semantic segmentation, CVPR 2023
> >
> > [4] Self-training with noisy student improves imagenet classification, CVPR 2020
> >
> > [5] Unbiased teacher for semi-supervised object detection. ICLR 2021

---

> > ### Comment · Reviewer_CdV8 · 2025-11-27
> >
> > Thank the authors for the response. I am aware that Self-training [4] was initially proposed for image classification. Meantime, it is one of the most basic practices in semi-supervised learning. Therefore, I do not think the improvement over such baseline is significant enough.

---

> > > ### Author Response · Authors · 2025-11-28
> > >
> > > The differences between CAST and [4] are not limited to the task discrepancy; the two papers pursue fundamentally different goals, setups, and training paradigms.
> > >
> > >
> > > [4] repeatedly retrains or enlarges a model with the goal of improving robustness. The authors describe their approach as self-training or knowledge expansion, and explicitly state in Section 2 that their method is not knowledge distillation. In contrast, CAST is a knowledge distillation framework designed to create a compact student expert by transferring VFM knowledge for pixel-level predictions, using contrastive learning and a stage wise training strategy.
> > >
> > >
> > > Furthermore, as noted earlier, a +0.5-1.0 AP improvement over strong VFM baselines is not considered negligible. Many prior works report gains of the same order of magnitude. For example, MI-DETR (CVPR’25), a VFM-enhanced detection method, reports a +0.6 AP improvement over the previous state of the art.
> > >
> > >
> > > ---
> > > ### Refrences
> > >
> > > Nan, Zhixiong, *et al.* “MI-DETR: An Object Detection Model with Multi-time Inquiries Mechanism.” *CVPR*, 2025.

---

> ### Author Response · Authors · 2025-11-27
>
> Dear Reviewer CdV8,
>
> Thank you again for your time and constructive feedback. We have carefully responded to all of your comments in detail. We understand that the discussion period is very busy, and if you have a moment, we would greatly appreciate any feedback on our responses.

---

### Official Review · Reviewer_xCYT · 2025-10-31

**Soundness:** 3
**Presentation:** 3
**Contribution:** 3
**Rating:** 6
**Confidence:** 3

**Summary:**

The paper presents CAST, a semi-supervised knowledge distillation framework to compress large VFMs for instance segmentation. It uses a three-stage pipeline: (1) refining the teacher model, (2) transferring knowledge to a compact student using a new instance-aware contrastive loss, and (3) fine-tuning the student. The resulting student model is 11x smaller in parameters but achieves superior performance compared to both its original teacher and other state-of-the-art methods.

**Strengths:**

1. Comprehensive Pipeline: CAST offers a well-structured SSKD pipeline that unifies teacher adaptation, knowledge distillation with a pixel-wise contrastive component, and student fine-tuning, systematically addressing the challenges of compressing large VFMs for instance segmentation.
2. Technical Rigor and Theoretical Insight: The paper details a mathematically sound instance-aware pixel-level contrastive loss, complete with negative sampling mechanisms incorporating fused mask-class cues.
3. Thorough Empirical Evaluation: The main results and ablation studies, along with figures robustly support claims of improved performance, efficiency, and bias reduction. The student notably improves over all relevant baselines and ablations.
4. Ablation and Diagnostic Depth: The paper reports meaningful ablations, including loss term contributions, the necessity of each pipeline stage, pixel-wise negative sampling strategies, hyperparameter sweeps, and architectural effects.

**Weaknesses:**

1. While the proposal is thoughtfully implemented and the contrastive calibration is well-integrated, the primary technical innovations seem incremental. The instance-aware contrastive loss largely adapts previous contrastive/self-supervised learning techniques with a new but straightforward mask and class score fusion for negative sampling.
2. The claims regarding “robustness” under low-label regimes stem from results on only two standard datasets. Broader generalization or domain robustness (e.g., in-the-wild images, transferred domains, or real edge-based applications) is not empirically validated, contrary to the aspiration mentioned in the introduction and conclusions.

**Questions:**

1. How robust is the instance-aware negative sampling scheme to highly imbalanced datasets, both between classes and between instances within a single image? Did you observe increased failure or co-located instance confusion under high crowding or many small objects?
2. What is the computational overhead of the contrastive loss (especially the instance-aware negative sampling) in practice relative to prior SSKD/contrastive frameworks?
3. Is the three-stage process of teacher adaptation, student distillation, and final fine-tuning critical? Would performance meaningfully degrade if adaptation and distillation are collapsed into a single objective?

---

> ### Author Response · Authors · 2025-11-21
>
> We appreciate the reviewer’s positive assessment of CAST’s overall design and contributions, as well as the constructive feedback on our work. We address all questions below.
> > **Q1:** How robust is the instance-aware negative sampling scheme to highly imbalanced datasets, both between classes and between instances within a single image? Did you observe increased failure or co-located instance confusion under high crowding or many small objects?
>
> **A1:** Our debiased instance-aware negative sampling remains stable throughout training. Empirically, we observe the false-negative rate stays consistently low, with sampling precision p = 1 − FNR > 0.9 (Fig. 3b).  This indicates that the vast majority of sampled negatives come from different instances, even in crowded scenes.
>
> We also find that the empirical inter-instance margin increases steadily with λpxl​ and remains stable across iterations (Fig. 3a), suggesting that features of nearby or small objects maintain reliable separability. We will provide additional qualitative examples in the revised PDF.
>
> we do note smaller gains on datasets with very large numbers of instance categories  (e.g., ADE20K with 100 instance classes vs. Cityscapes with 8), we attribute this to dataset-level class imbalance rather than instability of the sampling scheme itself.
>
>
> > **Q2:** What is the computational overhead of the contrastive loss (especially the instance-aware negative sampling) in practice relative to prior SSKD/contrastive frameworks?
>
> The additional overhead introduced by our contrastive loss and instance-aware negative sampling is modest. For example, On Cityscapes, a single Stage-2 student training run increases from ≈17 GPU hours to ≈21 GPU hours (24% overhead), with improving performance from 30.7 → 32.2maskAP.
>
> With this added KD time, the overall overhead remains comparable to prior SOTA semi-supervised frameworks. For example, we estimate the total cost relative to a strong baseline, Guided Distillation[1], using the same student encoder (DINOv2-S), identical batch size, and 10% labeled Cityscapes. Guided Distillation requires roughly 36 GPU hours (≈4 hrs Stage 1, ≈17 hrs Stage 2, ≈17 hrs Stage 3), whereas CAST requires roughly 32 GPU hours on the same setup (≈10 hrs Stage 1, ≈22 hrs Stage 2, <1 hr Stage 3).
>
>
> > **Q3:** Is the three-stage process of teacher adaptation, student distillation, and final fine-tuning critical? Would performance meaningfully degrade if adaptation and distillation are collapsed into a single objective?
>
> Table 3b provides a stage ablation demonstrating that removing the teacher adaptation stage leads to a substantial performance drop (33.9 → 25.7; −8.2 AP). This suggests that aligning the teacher to the target domain via pseudo-label refinement and contrastive calibration is critical for effective downstream distillation.
>
> Throughout our experimentation for designing the framework, we consistently found that a stage-wise learning procedure (teacher adaptation → student distillation → final fine-tuning), yielded the best final downstream performance. Based on this experience, we expect that merging the adaptation and distillation stages would still result in some degradation, though likely less severe than the “no-adaptation” setting and closer to the full pipeline’s performance. We plan to investigate this more thoroughly in future work.
>
> ---
> ### Refrences
>
> [1] Berrada, Tariq, et al. "Guided distillation for semi-supervised instance segmentation." Proceedings of the IEEE/CVF winter conference on applications of computer vision. 2024.

---

> > ### Comment · Reviewer_xCYT · 2025-11-25
> >
> > Thanks for your response. I will maintain my score and vote for acceptance.

---

### Official Review · Reviewer_AJrB · 2025-11-01

**Soundness:** 3
**Presentation:** 2
**Contribution:** 2
**Rating:** 4
**Confidence:** 4

**Summary:**

This paper introduces CAST, a semi-supervised knowledge distillation framework designed to compress large pre-trained vision foundation models (VFMs) into much smaller, efficient models. Experiments demonstrate the effectiveness of the method.

**Strengths:**

1. The organization of this paper is clear, which is easy to follow.

2. The experiments are good, and the ablation studies are comprehensive.

**Weaknesses:**

My main concern is mainly sourced from the insufficient discussion in related works.

(1) As the core design of the method is the instance-aware pixel-wise contrastive loss, there have been many contrastive learning based knowledge distillation methods. However, the paper lacks discussion on these previous works.

(2) Early works made a contrast between teachers’ and students’ features by employing different views generated from  using various samples’ features [1, 2] as well as gradients [3] stored in the memory buffer.

(3) In dense prediction tasks, G-DetKD [4] constructed ROI feature pairs and executed soft semantic-guided matching, which promoted the performance in object detection, and CIRKD [5] designed an implicit contrastive method that leveraged both pixel and region representations to learn structured information in the spatial dimension.  Furthermore, for pixel-level mimicking, Af-DCD [6] designs more fine-grained contrastive learning between pixels from the teacher and the student.

(4) Although they have different experimental settings, the paper is also in a related research line. Therefore, adding discussion to the related works, especially for segmentation tasks [4-6], is encouraged.


[1] Zhiyuan Fang, Jianfeng Wang, Lijuan Wang, Lei Zhang, Yezhou Yang, and Zicheng Liu. Seed:
Self-supervised distillation for visual representation. arXiv preprint arXiv:2101.04731, 2021

[2] Yonglong Tian, Dilip Krishnan, and Phillip Isola. Contrastive representation distillation. In
ICLR, 2020.

[3] Jinguo Zhu, Shixiang Tang, Dapeng Chen, Shijie Yu, Yakun Liu, Mingzhe Rong, Aijun Yang,
and Xiaohua Wang. Complementary relation contrastive distillation. In CVPR, 2021.

[4] Lewei Yao, Renjie Pi, Hang Xu, Wei Zhang, Zhenguo Li, and Tong Zhang. G-detkd: towards
general distillation framework for object detectors via contrastive and semantic-guided feature
imitation. In ICCV, 2021

[5] Chuanguang Yang, Helong Zhou, Zhulin An, Xue Jiang, Yongjun Xu, and Qian Zhang. Crossimage relational knowledge distillation for semantic segmentation. In CVPR, 2022.

[6]  Fan, Jiawei, et al. Augmentation-free dense contrastive knowledge distillation for efficient semantic segmentation. In NeurIPS 2023.

**Questions:**

See weakness

---

> ### Author Response · Authors · 2025-11-21
>
> > **Q: The reviewer notes that the paper lacks adequate discussion of prior contrastive KD methods. Specifically, they point out: (1) many existing contrastive KD works similar in spirit to ours, (2) early contrastive feature-matching approaches [1–3], and (3) dense-prediction methods for detection and segmentation [4–6]. They request deeper comparison and discussion in Related Work.**
>
> **A:**
>  We thank the reviewer for the constructive feedback and appreciate the positive comments on the clarity of our paper and the thoroughness of our experiments. Below, we address concerns regarding related work and clarify how our approach differs from prior methods. As all questions are relate to the coverage of related work, we address these points respectively below and clarify how our method differs from prior SOTA approaches.
>
>
> >**Contrastive Learning for Knowledge Distillation.**
>
>
> Early works such as CRD (ICLR’20) and SEED (ICLR’21) reformulate classical logit-based KD [Hinton et al., 2015] into a contrastive or self-supervised framework. CRD constructs positive pairs between teacher and student embeddings for the same image to maximize a lower bound of mutual information, thereby transferring structural information beyond softened logits. SEED extends this idea by matching the teacher’s similarity distribution over a memory queue in a self-supervised setting. CRCD (CVPR’21)  further enriches the contrastive signal by incorporating both feature relations and gradient relations stored in the memory buffer, capturing complementary structural and optimization-level information.
>
> >**Contrastive Distillation for Dense Prediction.**
>
> Subsequent work extends contrastive KD to dense tasks. G-DetKD (ICCV’21) introduces a semantic-guided feature imitation mechanism that contrasts teacher and student ROI features, enabling the student to mimic the teacher’s embeddings and semantic structure for object detection. CIRKD (CVPR’22) studies KD for semantic segmentation, motivated by the observation that existing approaches often overlook global semantic relations among pixels across images. CIRKD builds on SEED by introducing a shared memory bank that enables contrastive distillation between teacher and student feature maps at the pixel level. Af-DCD (NeurIPS’23) reduces the heavy memory requirements of CIRKD by contrasting teacher and student feature maps directly across both spatial and channel dimensions, eliminating the need for a memory queue. More recently, PCD (ICCV’23) proposes a SpatialAdaptor module to better align corresponding pixels between teacher and student output feature maps.
>
> >**Differences from Existing Works.**
>
> Existing methods have several limitations in our setting. **First**, they are not designed for the scale or architectural complexity of modern vision foundation models (VFMs). **Second**, they rarely address instance segmentation. **Third**, most prior work assumes a single teacher student pair (e.g., ResNet-101 → ResNet-50) and therefore does not consider the more challenging case of using an ensemble of teachers, where there is no straightforward, shared feature map from which to distill. **Finally**, an important practical distinction is that many existing approaches rely on contrastive objectives aimed at aligning teacher and student feature spaces. However, recent work shows that distillation based solely on feature- or logit-level alignment can underperform compared with task-aware, pseudo or label driven distillation strategies [Vemulapalli et al., ICML'24]. In contrast, we do not perform teacher–student contrastive learning; we use contrastive learning only as a self-supervised mechanism on unlabeled data to enhance the model's structural consistency.
>
> >**Our Contributions Relative to Prior Art.**
>
> To address these limitations, we design a novel semi-supervised KD framework tailored for VFMs distillation and dense prediction tasks. As part of this setup contrastive learning is used to explore **1.** whether contrastive learning can additionally leverage the representations of large amounts of unlabeled data in a self-supervised manner to strengthen the student’s structural consistency **2.** whether this additional self-supervised signal can be synergized with supervised and semi-supervised losses[L137].
>
> >**Further related work and updates**
>
> Lastly, we note that lines L131–L140 of the Related Work section summarize earlier contrastive learning frameworks spanning both global and dense formulations. We will further update  the manuscript to incorporate these new citations and refine the Related Work section accordingly.
>
>
> > **Due to space limitations, we provide the reference list in the follow-up comment.**

---

> > ### Author Response · Authors · 2025-11-21
> >
> > ---
> >
> > ### References
> >
> > [1] Zhiyuan Fang, Jianfeng Wang, Lijuan Wang, Lei Zhang, Yezhou Yang, and Zicheng Liu. Seed: Self-supervised distillation for visual representation. arXiv preprint arXiv:2101.04731, 2021
> >
> > [2] Yonglong Tian, Dilip Krishnan, and Phillip Isola. Contrastive representation distillation. In ICLR, 2020.
> >
> > [3] Jinguo Zhu, Shixiang Tang, Dapeng Chen, Shijie Yu, Yakun Liu, Mingzhe Rong, Aijun Yang, and Xiaohua Wang. Complementary relation contrastive distillation. In CVPR, 2021.
> >
> > [4] Lewei Yao, Renjie Pi, Hang Xu, Wei Zhang, Zhenguo Li, and Tong Zhang. G-detkd: towards general distillation framework for object detectors via contrastive and semantic-guided feature imitation. In ICCV, 2021
> >
> > [5] Chuanguang Yang, Helong Zhou, Zhulin An, Xue Jiang, Yongjun Xu, and Qian Zhang. Crossimage relational knowledge distillation for semantic segmentation. In CVPR, 2022.
> >
> > [6] Fan, Jiawei, et al. Augmentation-free dense contrastive knowledge distillation for efficient semantic segmentation. In NeurIPS 2023.
> >
> > [7] Huang, Junqiang, and Zichao Guo. Pixel-wise contrastive distillation. In CVPR 2023
> >
> > [8] Geoffrey Hinton, Oriol Vinyals, and Jeff Dean. Distilling the knowledge in a neural network. arXiv preprint arXiv:1503.02531, 2015.
> >
> > [9] Vemulapalli, Raviteja, et al. Knowledge transfer from vision foundation models for efficient training of small task-specific models. In ICML 2024

---

> ### Author Response · Authors · 2025-11-27
>
> Dear Reviewer AJrB,
>
> Thank you again for your time and constructive feedback. We have carefully responded to all of your comments in detail. We understand that the discussion period is very busy, and if you have a moment, we would greatly appreciate any feedback on our responses.

---

### Author Response · Authors · 2025-12-03
**Concluding Remarks**

We sincerely thank all reviewers for their valuable comments and constructive suggestions. To facilitate the discussion among the reviewers and the area chair, we have summarized the reviewers’ feedback in the table below.


| Strengths                                                                 | AJrB | xCYT | CdV8 | anhQ |
|----------------------|:----:|:----:|:----:|:----:|
| Clear and well-organized paper                         | ✔️   |      | ✔️   |      |
| Comprehensive experiments and ablations        | ✔️   |      |      | ✔️   |
| Strong, well-structured SSKD pipeline                                     |      | ✔️   |      |      |
| Technical rigor and theoretical insight                                   |      | ✔️   |      | ✔️   |
| Good motivation for unlabeled data                                                          |      |      | ✔️   |   ✔️   |
| First unified VFM adaptation, dense contrastive learning, and extreme compression  |      |      |      | ✔️  |
| Novel instance-aware negative sampler                                     |      |      |      | ✔️   |
| Failure modes analyzed and mitigated                                      |      |      |      | ✔️   |
|
| **Weaknesses**
| Insufficient discussion of contrastive KD works                   | ✔️   |      |      |      |
| Contrastive learning signal perceived as incremental                      |      | ✔️   | ✔️   |      |
| Limited robustness validation|      | ✔️   |      |      |
| Small unlabeled-data gains|      |      | ✔️   |      |
| Missing single teacher gain breakdown|      |      |      | ✔️   |
| **Rating**                                                                 | 4    | 6    | 4    | 6    |
| **Confidence**                                                             | 4    | 3    | 3    | 5    |

We appreciate the reviewers’ recognition of the strengths of our work,  including the clarity of the paper, the technical soundness of the proposed method, the comprehensiveness of our ablations, and the effectiveness of CAST in compressing large vision foundation models(VFMs) for instance segmentation.

To address the reviewers' comments on the identified limitations, we provided additional results and clarifications as follows:

1. **Insufficient discussion of prior contrastive KD works (AJrB)**: We discussed the contrastive KD literature noted by the reviewer, added additional relevant works, and clarified the differences from prior SOTA methods. We also incorporated a summary of this discussion into the manuscript.
2.  **Prior works on dense contrastive learning (CdV8)**: We highlighted the differences between prior methods and our use of contrastive learning as a self-supervised signal to capture the underlying structure of unlabeled data within our broader semi-supervised distillation pipeline.
3. **Marginal gains from unlabeled images (CdV8)**: We directed the reviewer to Table 1 to illustrate the gains achieved across the full pipeline and noted that, due to strong VFM baseline performance, the ordering of VFM gains can appear modest in isolation. However, our results demonstrate that unlabeled data provides substantial benefits when used across the full pipeline. We elaborated the rationale for teacher adaptation and highlighted the empirical advantage of our approach over self-training baselines.
4. **Empirical advantage over self-training method [4] (CdV8)**: We included the differences between Noisy Student [4] and CAST in terms of task and setup.

Furthermore, we addressed additional questions raised by the reviewers:

1. **Robustness of the instance-aware negative sampling scheme in crowded or imbalanced classes (xCYT)**: We showed that our debiased instance-aware negative sampling remains stable even in highly crowded settings, with consistently low false-negative rates (precision > 0.9) and stable inter-instance margins (Fig. 3a - 3b). While datasets with larger numbers of class categories (e.g., ADE20K) yield smaller overall gains, which is attributed to dataset-level class imbalance rather than instability in the sampling scheme itself.
2. **Computational overhead of the contrastive loss and comparison with prior SOTA (xCYT)**:  We evaluated the computational overhead of the contrastive loss in isolation and compared the overall pipeline efficiency against a prior SOTA method.
3. **Three stage process (xCYT)**: Building on the stage ablation in Table 2b, we explained the necessity of each stage and the benefits of stage-wise training.

**Additional Updates**:
- We added a new paragraph in the manuscript (L99) summarizing the prior works discussion with reviewer AJrB.
- We added Figure 8 (Appendix G), with additional qualitative examples.

We sincerely thank all reviewers and the area chair for their time, patience, and thoughtful feedback.

---

### Meta-Review · Area_Chair_zUEx · 2026-01-06

**Summary:**

The reviewer anhQ raised a concern that the paper fuses Grounding-DINO + SAM-2 as the teacher, without showing results from single teacher distillation. This makes it unclear whether the reported gains come from the contrastive loss or from ensembling complementary models. The authors’ response acknowledges this issue and states that they plan to disentangle the effects of each teacher, but no results or analyses are currently provided. As such, the concern remains unaddressed in the current submission and is deferred to future work. However, this is an important issue that must be addressed in the paper rather than left for future work.

In addition, two reviewers raised questions regarding the novelty of the proposed method. This raises a concern about the novelty of the proposed method, as the work needs to be compared and discussed with more relevant prior works to better contextualise and highlight its unique contributions.

Given these two major issues, the AC believes the paper is not ready for publication in its current form.

**Reviewer Concerns:**

Most concerns have been addressed except the below major ones

The proposed method fuses Grounding-DINO + SAM-2 as the teacher, without showing results from single teacher distillation. This makes it unclear whether the reported gains come from the contrastive loss or from ensembling complementary models. The authors’ response acknowledges this issue and states that they plan to disentangle the effects of each teacher, but no results or analyses are currently provided. As such, the concern remains unaddressed in the current submission and is deferred to future work. However, this is an important issue that must be addressed in the paper rather than left for future work.

In addition, two reviewers raised questions regarding the novelty of the proposed method. This raises a concern about the novelty of the proposed method, as the work needs to be compared and discussed with more relevant prior works to better contextualise and highlight its unique contributions. While the rebuttal tries to address this issue, however, the AC finds that the novelty of the proposed method remains insufficiently contextualised.

**Reviewer Scores:**

I think the quality of reviews aren't good enough. For example, Reviewer AJrB only asked for adding several papers to the related work section and no other justification for their score (i.e., 6). It only says easy to read and good experiments.

Another example is the final feedback from the reviewer anhQ. This reviewer asked for showing results from single teacher distillation as it is unclear whether the reported gains come from the contrastive loss or from ensembling complementary models. However, the rebuttal acknowledges this issue and states that they plan to disentangle the effects of each teacher, but no results or analyses are currently provided. The reviewer then noted that they keep their score (i.e., 6).

---

### Decision · Program_Chairs · 2026-01-26

Reject